# Knowledge Distillation by On-the-Fly Native Ensemble

**Xu Lan**[1], **Xiatian Zhu**[2], and **Shaogang Gong**[1]

[1]Queen Mary University of London
[2]Vision Semantics Limited

## Abstract

Knowledge distillation is effective to train the small and generalisable network models for meeting the low-memory and fast running requirements. Existing offline distillation methods rely on a strong pre-trained teacher, which enables favourable knowledge discovery and transfer but requires a complex two-phase training procedure. Online counterparts address this limitation at the price of lacking a high-capacity teacher. In this work, we present an On-the-fly Native Ensemble (ONE) learning strategy for one-stage online distillation. Specifically, ONE only trains a single multi-branch network while simultaneously establishing a strong teacher on-the-fly to enhance the learning of target network. Extensive evaluations show that ONE improves the generalisation performance of a variety of deep neural networks more significantly than alternative methods on four image classification dataset: CIFAR10, CIFAR100, SVHN, and ImageNet, whilst having the computational efficiency advantages.

## 1 Introduction

Deep neural networks have gained impressive success in many computer vision tasks [1, 2, 3, 4, 5, 6, 7, 8]. However, the performance advantages often come at the cost of training and deploying resource-intensive networks with large depth and/or width [9, 4, 2]. This leads to the necessity of developing compact yet discriminative models. Knowledge distillation [10] is one general meta-solution among the others such as parameter binarisation [11, 12] and filter pruning [13]. The distillation process begins with training a high-capacity *teacher* model (or an ensemble of models), followed by learning a smaller *student* model which is encouraged to match the teacher's predictions [10] or feature representations [14, 15]. While promising the student model quality improvement from aligning with a pre-trained teacher model, this strategy requires a longer training process, significant extra computational cost and large memory (for a heavy teacher) in a more complex multi-phase training procedure. These are commercially unattractive [16].

To simplify the distillation training process as above, simultaneous distillation algorithms [17, 16] have been developed to perform knowledge online teaching in a one-phase training procedure. Instead of pre-training a static teacher model, these methods train simultaneously a set of (typically two) student models which learn from each other in a peer-teaching manner. This approach merges the training processes of the teacher and student models, and uses the peer network to provide the teaching knowledge. Beyond the original understanding of distillation that requires the teacher model larger than the student, this online distilling strategy can improve the performance of any-capacity models, leading to a more generically applicable technique. Such a peer-teaching strategy sometimes even outperforms the teacher based offline distillation. The plausible reason is that the large teacher model tends to overfit the training data therefore leading to less extra knowledge on top of the manually labelled annotations [16].

However, the existing online distillation methods have a number of drawbacks: (1) Each peer-student model may only provide limited extra information, resulting in suboptimal distillation; (2) Training multiple students causes a significant increase of computational cost and resource burdens; (3) They require asynchronous model updating which has a notorious need of carefully ordering the operations of label prediction and gradient back-propagation across networks. We consider that all the weaknesses are due to the lack of an appropriate teacher role in the online distillation processing.

In this work, we propose a novel online knowledge distillation method that is not only more efficient (lower training cost) but also more effective (higher model generalisation improvement) as compared to previous alternative methods. In *training*, the proposed approach constructs a multi-branch variant of a given target network by adding auxiliary branches, creates a native ensemble teacher model from all branches on-the-fly, and learns simultaneously each branch plus the teacher model subject to the same target label constraints. Each branch is trained with two objective loss terms: a conventional softmax cross-entropy loss which matches with the ground-truth label distributions, and a distillation loss which aligns to the teacher's prediction distributions. Comparing with creating a set of student networks, a multi-branch *single* model is more efficient to train whilst achieving superior generalisation performance and avoiding asynchronous model update. In *test*, we simply convert the trained multi-branch model back to the original (single-branch) network architecture by removing the auxiliary branches, therefore *not* increasing test-time cost. In doing so, we derive an **On-the-Fly Native Ensemble** (ONE) teacher based simultaneous distillation training approach that not only eliminates the computationally expensive need for pre-training the teacher model in an isolated stage as the offline counterparts, but also further improves the quality of online distillation.

Extensive experiments on four benchmarks (CIFAR10/100, SVHN, and ImageNet) show that the proposed ONE distillation method enables to train more generalisable target models in an one-phase process than the alternative strategies of offline learning a larger teacher network or simultaneously distilling peer students, the previous state-of-the-art techniques for training small target models.

## 2 Related Work

**Knowledge Distillation.** There are existing attempts at knowledge transfer between varying-capacity network models [18, 10, 14, 15]. Hinton et al. [10] distilled knowledge from a large teacher model to improve a small target net. The rationale behind is taking advantage of extra supervision provided by the teacher model during training the target model, beyond a conventional supervised learning objective such as the cross-entropy loss subject to the training data labels. The extra supervision is typically extracted from a pre-trained powerful teacher model in form of class posterior probabilities [10], feature representations [14, 15], or inter-layer flow (the inner product of feature maps) [19].

Recently, knowledge distillation has been exploited to distil easy-to-train large networks into harder-to-train small networks [15], to transfer knowledge within the same network [20, 21], and to transfer high-level semantics across layers [8]. Earlier distillation methods often take an offline learning strategy, requiring at least two phases of training. The more recently proposed deep mutual learning [17] overcomes this limitation by conducting an online distillation in one-phase training between two peer student models. Anil et al. [16] further extended this idea to accelerate the training of large scale distributed neural networks.

However, the existing online distillation methods lack a strong "teacher" model which limits the efficacy of knowledge discovery. As the offline counterpart, multiple nets are needed to be trained therefore computationally expensive. We overcome both limitations by designing a new online distillation training algorithm characterised by simultaneously learning a teacher on-the-fly and the target net as well as performing batch-wise knowledge transfer in an one-phase training procedure.

**Multi-branch Architectures.** Multi-branch based neural networks have been widely exploited in computer vision tasks [3, 22, 4]. For example, ResNet [4] can be thought of as a category of two-branch networks where one branch is the identity mapping. Recently, "grouped convolution" [23, 24] has been used as a replacement of standard convolution in constructing multi-branch net architectures. These building blocks are usually utilised as templates to build deeper networks to gain stronger model capacities. Despite sharing the multi-branch principle, our ONE method is fundamentally different from such existing methods since our objective is to improve the training quality of any target network, but *not* to propose a new multi-branch building block. In other words, our method is a meta network learning algorithm, independent of the network architecture design.

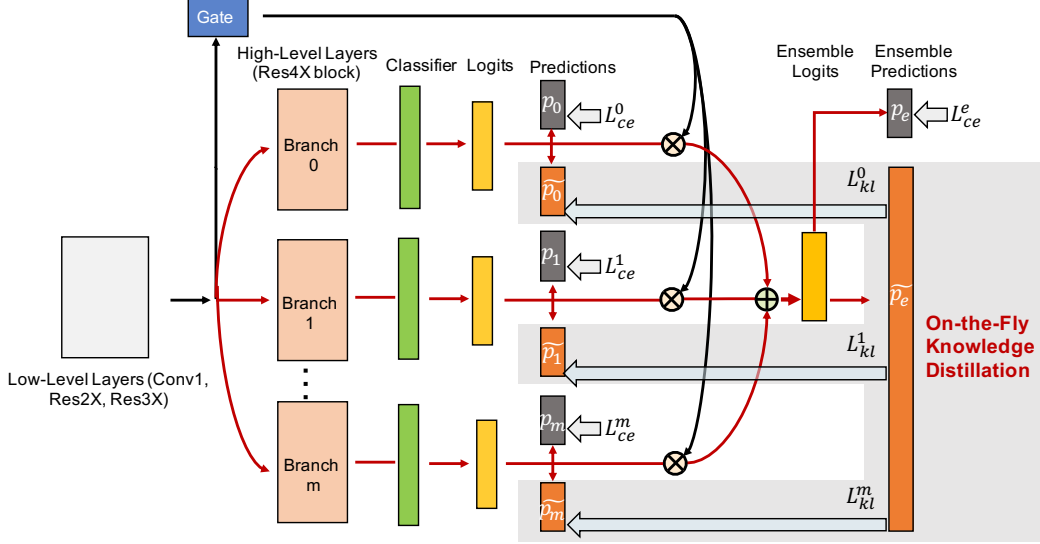

Figure 1: Overview of online distillation training of ResNet-110 by the proposed On-the-fly Native Ensemble (ONE). With ONE, we start by reconfiguring the target network by adding $m$ auxiliary branches on shared low-level layers. All branches together with shared layers make individual models, all of which are then used to construct a stronger teacher model. During the mini-batch training process, we employ the teacher to assemble knowledge of branch models on-the-fly, which is in turn distilled back to all branches to enhance the model learning in a closed-loop form. In test, auxiliary branches are discarded or kept according to the deployment efficiency requirement.

## 3 Knowledge Distillation by On-the-Fly Native Ensemble

We formulate an online distillation training method based on a concept of On-the-fly Native Ensemble (ONE). For understanding convenience, we take ResNet-110 [4] as an example. It is straightforward to apply ONE to other network architectures. For model training, we often have access to $n$ labelled training samples $\mathcal{D} = \{(\boldsymbol{x}_i, y_i)\}_i^n$ with each belonging to one of $C$ classes $y_i \in \mathcal{Y} = \{1, 2, \cdots, C\}$. The network $\boldsymbol{\theta}$ outputs a probabilistic class posterior $p(c|\boldsymbol{x}, \boldsymbol{\theta})$ for a sample $\boldsymbol{x}$ over a class $c$ as:

$$p(c|\boldsymbol{x}, \boldsymbol{\theta}) = f_{sm}(\boldsymbol{z}) = \frac{\exp(\boldsymbol{z}^c)}{\sum_{j=1}^{C} \exp(\boldsymbol{z}^j)}, \quad c \in \mathcal{Y} \tag{1}$$

where $\boldsymbol{z}$ is the logits or unnormalised log probability outputted by the network $\boldsymbol{\theta}$. To train a multi-class classification model, we typically adopt the Cross-Entropy (CE) measurement between the predicted and ground-truth label distributions as the objective loss function:

$$\mathcal{L}_{\text{ce}} = -\sum_{c=1}^{C} \delta_{c,y} \log\left(p(c|\boldsymbol{x}, \boldsymbol{\theta})\right) \tag{2}$$

where $\delta_{c,y}$ is Dirac delta which returns 1 if $c$ is the ground-truth label, and 0 otherwise. With the CE loss, the network is trained to predict the correct class label in a principle of maximum likelihood. To further enhance the model generalisation, we concurrently distil extra knowledge from an on-the-fly native ensemble (ONE) teacher to each branch in training.

**On-the-Fly Native Ensemble.** An overview of the ONE architecture is depicted in Fig 1. The ONE consists of two components: **(1)** $m$ auxiliary branches with the same configuration (Res4X block and an individual classifier), each of which serves as an independent classification model with shared low-level stages/layers. This is because low-level features are largely shared across different network instances and sharing them allows to reduce the training cost. **(2)** A gating component which learns to ensemble all $(m+1)$ branches to build a stronger teacher model. It is constructed by one fully connected (FC) layer followed by batch normalisation, ReLU activation, and softmax, using the same input features as the branches.

Our ONE method is established based on a multi-branch design specially for model training with several merits: (1) Enable the possibility of creating a strong teacher model without training a set of networks at a high computational cost; (2) Introduce a multi-branch simultaneous learning regularisation which benefits model generalisation (Fig 3(a)); (3) Avoid the tedious need for asynchronous update between multiple networks.

Under the reconfiguration of network, we add a separate CE loss $\mathcal{L}_{ce}^i$ to each branch which simultaneously learns to predict the same ground-truth class label of a training sample. While sharing the most layers, each branch can be considered as an independent multi-class classifier in that all of them independently learn high-level semantic representations. Consequently, taking the ensemble of all branches (classifiers) can make a stronger teacher model. One common way of ensembling models is to average individual predictions. This may ignore the diversity and importance variety of the member models of an ensemble. We therefore learn to ensemble by a gating component as:

$$\boldsymbol{z}_e = \sum_{i=0}^{m} g_i \cdot \boldsymbol{z}_i \tag{3}$$

where $g_i$ is the importance score of the $i$-th branch's logits $\boldsymbol{z}_i$, and $\boldsymbol{z}_e$ is the logits of the ONE teacher. In particular, we denote the original branch as $i = 0$ for indexing convenience. We train the ONE teacher model with the CE loss $\mathcal{L}_{ce}^e$ (Eq (2)) the same as the branches.

**Knowledge Distillation.** Given the teacher's logits of each training sample, we distil this knowledge back into all branches in a closed-loop form. For facilitating knowledge transfer, we compute soft probability distributions at a temperature of $T$ for individual branches and the ONE teacher as:

$$\tilde{p}_i(c|\boldsymbol{x}, \boldsymbol{\theta}^i) = \frac{\exp(\boldsymbol{z}_i^c/T)}{\sum_{j=1}^{C} \exp(\boldsymbol{z}_i^j/T)}, c \in \mathcal{Y} \tag{4}$$

$$\tilde{p}_e(c|\boldsymbol{x}, \boldsymbol{\theta}^e) = \frac{\exp(\boldsymbol{z}_e^c/T)}{\sum_{j=1}^{C} \exp(\boldsymbol{z}_e^j/T)}, c \in \mathcal{Y} \tag{5}$$

where $i$ denotes the branch index, $i = 0, \cdots, m$, $\boldsymbol{\theta}^i$ and $\boldsymbol{\theta}^e$ the parameters of the branch and teacher models respectively. Higher values of $T$ lead to more softened distributions.

To quantify the alignment between individual branches and the teacher in their predictions, we use the Kullback Leibler divergence from branches to the teacher written as:

$$\mathcal{L}_{kl} = \sum_{i=0}^{m} \sum_{j=1}^{C} \tilde{p}_e(j|\boldsymbol{x}, \boldsymbol{\theta}^e) \log \frac{\tilde{p}_e(j|\boldsymbol{x}, \boldsymbol{\theta}^e)}{\tilde{p}_i(j|\boldsymbol{x}, \boldsymbol{\theta}^i)}. \tag{6}$$

**Overall Loss Function.** We obtain the overall loss function for online distillation training by the proposed ONE as:

$$\mathcal{L} = \sum_{i=0}^{m} \mathcal{L}_{ce}^i + \mathcal{L}_{ce}^e + T^2 * \mathcal{L}_{kl} \tag{7}$$

where $\mathcal{L}_{ce}^i$ and $\mathcal{L}_{ce}^e$ are the conventional CE loss terms associated with the $i$-th branch and the ONE teacher, respectively. The gradient magnitudes produced by the soft targets $\tilde{p}$ are scaled by $\frac{1}{T^2}$, so we multiply the distillation loss term by a factor $T^2$ to ensure that the relative contributions of ground-truth and teacher probability distributions remain roughly unchanged. Note, the entire ONE objective function of ONE is *not* an ensemble learning since (1) these loss functions corresponding to the models with different roles, and (2) the conventional ensemble learning often takes *independent* training of member models.

**Model Training and Deployment.** The model optimisation and deployment details are summarised in Alg 1. Unlike the two-phase offline distillation training, the target network and the ONE teacher are trained simultaneously and collaboratively, with the knowledge distillation from the teacher to the target being conducted in each mini-batch and throughout the whole training procedure. Since there is one multi-branch network rather than multiple networks, we only need to carry out the same stochastic gradient descent through $(m + 1)$ branches, and training the whole network until

**Algorithm 1** Knowledge Distillation by On-the-Fly Native Ensemble

---

1: **Input**: Labelled training data $\mathcal{D}$; Training epoch number $\tau$; Auxiliary branch number $m$;
2: **Output**: Trained target CNN model $\boldsymbol{\theta}^0$, and auxiliary models $\{\boldsymbol{\theta}^i\}_{i=1}^m$;
3: **/* Training */**
4: **Initialisation**: t=1; Randomly initialise $\{\boldsymbol{\theta}^i\}_{i=0}^m$;
5: **while** $t \leq \tau$ **do**
6:       Compute predictions of all individual branches $\{p_i\}_{i=0}^m$ (Eq (1));
7:       Compute the teacher logits (Eq (3));
8:       Compute the soft targets of all the branch and teacher models (Eq (4));
9:       Distil knowledge from the teacher back to all the branch models (Eq (6));
10:      Compute the final ONE loss function (Eq (7));
11:      Update the model parameters $\{\boldsymbol{\theta}^i\}_{i=0}^m$ by a SGD algorithm.
12: **end**
13: **/* Testing */**
14: **Single model deployment:** Use $\boldsymbol{\theta}^0$;
15: **Ensemble deployment (ONE-E):** Use $\{\boldsymbol{\theta}^i\}_{i=0}^m$.

---

converging, as the standard single-model incremental batch-wise training. There is no complexity of asynchronously updating among different networks which is required in deep mutual learning [17].

Once the model is trained, we simply remove all the auxiliary branches and obtain the original network architecture for deployment. Hence, our ONE method does not increase the test-time cost. However, if there is less constraint on the computation budget and the model performance is more important, we can deploy it as an ensemble model with all trained branches, denoted as "ONE-E".

## 4 Experiments

**Datasets.** We used four multi-class categorisation benchmark datasets in our evaluations (Fig 2). **(1)** *CIFAR10* [25]: A natural images dataset that contains 50,000/10,000 training/test samples drawn from 10 object classes (in total 60,000 images). Each class has 6,000 images sized at $32 \times 32$ pixels. **(2)** *CIFAR100* [25]: A similar dataset as CIFAR10 that also contains 50,000/10,000 training/test images but covering 100 fine-grained classes. Each class has 600 images. **(3)** *SVHN*: The Street View House Numbers (SVHN) dataset consists of 73,257/26,032 standard training/text images and an extra set of 531,131 training images. We used all the training data *without* using data augmentation as [26, 27]. **(4)** *ImageNet*: The 1,000-class dataset from ILSVRC 2012 [28] provides 1.2 million images for training, and 50,000 for validation.

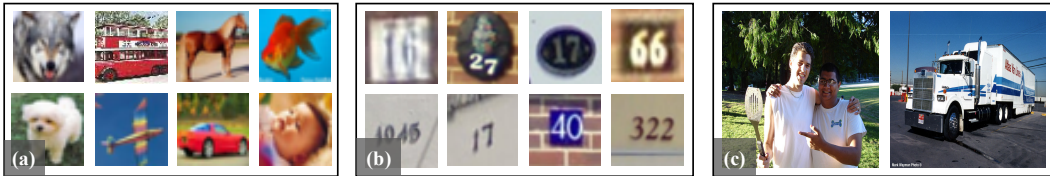

Figure 2: Example images from (a) CIFAR, (b) SVHN, and (c) ImageNet.

**Performance Metrics.** We adopted the common top-$n$ ($n$=1, 5) classification error rate. To measure the computational cost of model training and test, we used the criterion of floating point operations (FLOPs). For any network trained by ONE, we reported the average performance of all branch outputs with standard deviation.

**Experiment Setup.** We implemented all networks and model training procedures in Pytorch. For all datasets, we adopted the same experimental settings as [29, 23] for making fair comparisons. We used the SGD with Nesterov momentum and set the momentum to 0.9. We deployed a standard learning rate schedule that drops from 0.1 to 0.01 at 50% training and to 0.001 at 75%. For the training budget, we set 300/40/90 epochs for CIFAR/SVHN/ImageNet, respectively. We adopted a

| Method | CIFAR10 | CIFAR100 | SVHN | Params |
|---|---|---|---|---|
| ResNet-32 [4] | 6.93 | 31.18 | 2.11 | 0.5M |
| ResNet-32 + **ONE** | **5.99±0.05** | **26.61±0.06** | **1.83±0.05** | 0.5M |
| ResNet-110 [4] | 5.56 | 25.33 | 2.00 | 1.7M |
| ResNet-110 + **ONE** | **5.17±0.07** | **21.62±0.26** | **1.76±0.07** | 1.7M |
| ResNeXt-29($8\times64d$) [23] | 3.69 | 17.77 | 1.83 | 34.4M |
| ResNeXt-29($8\times64d$) + **ONE** | **3.45±0.04** | **16.07±0.08** | **1.70±0.03** | 34.4M |
| DenseNet-BC(L=190, k=40) [30] | 3.32 | 17.53 | 1.73 | 25.6M |
| DenseNet-BC(L=190, k=40) + **ONE** | **3.13±0.07** | **16.35±0.05** | **1.63±0.05** | 25.6M |

Table 1: Evaluation of our ONE method on CIFAR and SVHN. Metric: Error rate (%).

3-branch ONE ($m\!=\!2$) design unless stated otherwise. We separated the last block of each backbone net from the parameter sharing (except on ImageNet we separated the last 2 blocks to give more learning capacity to branches) without extra structural optimisation (see ResNet-110 for example in Fig 1). Following [10], we set $T = 3$ in all the experiments. Cross-validation of this parameter $T$ may give better performance but at the cost of extra model tuning.

## 4.1 Evaluation of On-the-Fly Native Ensemble

**Results on CIFAR and SVHN.** Table 1 compares top-1 error rate performances of four varying-capacity state-of-the-art network models trained by the conventional and our ONE learning algorithms. We have these observations: (1) All different networks benefit from the ONE training algorithm, particularly with small models achieving larger performance gains. This suggests a generic superiority of our method for online knowledge distillation from the on-the-fly teacher to the target student model. (2) All individual branches have similar performances, indicating that they have made sufficient agreement and exchanged respective knowledge to each other well through the proposed ONE teacher model during training.

| Method | Top-1 | Top-5 |
|---|---|---|
| ResNet-18 [4] | 30.48 | 10.98 |
| ResNet-18 + **ONE** | **29.45±0.23** | **10.41±0.12** |
| ResNeXt-50 [23] | 22.62 | 6.29 |
| ResNeXt-50 + **ONE** | **21.85±0.07** | **5.90±0.05** |
| SeNet-ResNet-18 [31] | 29.85 | 10.72 |
| SeNet-ResNet-18 + **ONE** | **29.02±0.17** | **10.13±0.12** |

Table 2: Evaluation of our ONE method on ImageNet. Metric: Error rate (%).

**Results on ImageNet.** Table 2 shows the comparative performances on the 1000-classes ImageNet. It is shown that the proposed ONE learning algorithm again yields more effective training and more generalisable models in comparison to the vanilla SGD. This indicates that our method is generically applicable in large scale image classification settings.

| Target Network | ResNet-32 | | | ResNet-110 | | |
|---|---|---|---|---|---|---|
| Metric | Error (%) | TrCost | TeCost | Error (%) | TrCost | TeCost |
| KD [10] | <span style="color:blue">**28.83**</span> | 6.43 | 1.38 | N/A | N/A | N/A |
| DML [17] | 29.03±0.22* | <span style="color:blue">**2.76**</span> | 1.38 | <span style="color:blue">**24.10±0.72**</span> | <span style="color:blue">**10.10**</span> | 5.05 |
| **ONE** | <span style="color:red">**26.61±0.06**</span> | <span style="color:red">**2.28**</span> | 1.38 | <span style="color:red">**21.62±0.26**</span> | <span style="color:red">**8.29**</span> | 5.05 |

Table 3: Comparison with knowledge distillation methods on CIFAR100. "*": Reported results. TrCost/TeCost: Training/test cost, in unit of $10^8$ FLOPs. <span style="color:red">**Red**</span>/<span style="color:blue">**Blue**</span>: Best and second best results.

| Network | ResNet-32 | | | ResNet-110 | | |
|---|---|---|---|---|---|---|
| Metric | Error (%) | TrCost | TeCost | Error (%) | TrCost | TeCost |
| Snapshot Ensemble [32] | 27.12 | **1.38** | 6.90 | 23.09* | **5.05** | 25.25 |
| 2-Net Ensemble | 26.75 | 2.76 | **2.76** | 22.47 | 10.10 | **10.10** |
| 3-Net Ensemble | **25.14** | 4.14 | 4.14 | **21.25** | 15.15 | 15.15 |
| **ONE-E** | **24.63** | **2.28** | **2.28** | **21.03** | **8.29** | **8.29** |
| **ONE** | 26.61 | 2.28 | 1.38 | 21.62 | 8.29 | 5.05 |

Table 4: Comparison with ensembling methods on CIFAR100. "*": Reported results. TrCost/TeCost: Training/test cost, in unit of $10^8$ FLOPs. **Red**/**Blue**: Best and second best results.

## 4.2 Comparison with Distillation Methods

We compared our ONE method with two representative distillation methods: Knowledge Distillation (KD) [10] and Deep Mutual Learning (DML) [17]. For the offline competitor KD, we used a large network ResNet-110 as the teacher and a small network ResNet-32 as the student. For the online methods DML and ONE, we evaluated their performances using either ResNet-32 or ResNet-110 as the target student model. We observed from Table 3 that: (1) ONE outperforms both KD (offline) and DML (online) distillation methods in error rate, validating the performance advantages of our method over alternative algorithms when applied to different CNN models. (2) ONE takes the least model training cost and the same test cost as others, therefore giving the most cost-effective solution.

## 4.3 Comparison with Ensembling Methods

Table 4 compares the performances of our multi-branch (3 branches) based model ONE-E and standard ensembling methods. It is shown that ONE-E yields not only the best test error but also enables most efficient deployment with the lowest test cost. These advantages are achieved at the second lowest training cost. Whilst Snapshot Ensemble takes the least training cost, its generalisation capability is unsatisfied with a notorious drawback of much higher deployment cost.

It is worth noting that ONE (without branch ensemble) already outperforms comprehensively a 2-Net Ensemble in terms of error rate, training and test cost. Comparing a 3-Net Ensemble, ONE approaches the generalisation capability whilst having larger model training and test efficiency advantages.

| Configuration | Full | W/O Online Distillation | W/O Sharing Layers | W/O Gating |
|---|---|---|---|---|
| ONE | **21.62±0.26** | 24.73±0.20 | 22.45±0.52 | 22.26±0.23 |
| ONE-E | 21.03 | 21.84 | **20.57** | 21.79 |

Table 5: Model component analysis on CIFAR100. Network: ResNet-110.

## 4.4 Model Component Analysis

Table 5 shows the benefits of individual ONE components on CIFAR100 using ResNet-110. We have these observations: (1) **Without online distillation** (Eq (6)), the target network suffers a performance drop of 3.11% (24.73-21.62) in test error rate. This performance drop validates the efficacy and quality of the ONE teacher in terms of performance superiority over individual branch models. This can be more clearly seen in Fig 3 that the ONE teacher fits better to training data and generalises better to test data. Due to the closed-loop design, the ONE teacher also mutually benefits from distillation, reducing its error rate from 21.84% to 21.03%. With distillation, the target model effectively approaches the ONE teacher (Fig 3(a) vs 3(b)) on both training and test error performance, indicating the success of teacher knowledge transfer. Interestingly, even without distillation, ONE still achieves better generalisation than the vanilla algorithm. This suggests that our multi-branch design brings some positive regularisation effect by concurrently and jointly learning the shared low-level layers subject to more diverse high-level representation knowledge. (2) **Without sharing the low-level layers** not only increases the training cost (83% increase), but also leads to weaker performance (0.83% error rate increase). The plausible reason is a lack of multi-branch regularisation effect as indicated in Fig 3(a). (3) Using average ensemble of branches **without gating** (Eq (3)) causes a

| Branch # | 1 | 2 | 3 | 4 | 5 |
|---|---|---|---|---|---|
| Error (%) | 31.18 | 27.38 | 26.68 | 26.58 | **26.52** |

Table 6: Benefit of adding branches to ONE on CIFAR100. Network: ResNet-32.

performance decrease of $0.64\%(22.26\text{-}21.62)$. This suggests the benefit of adaptively exploiting the branch diversity in forming the ONE teacher.

The main experiments use 3 branches in ONE. Table 6 shows that ONE scales well with more branches and the ResNet-32 model generalisation improves on CIFAR100 with the number of branches added during training hence its performance advantage over the independently trained network ($31.18\%$ error rate).

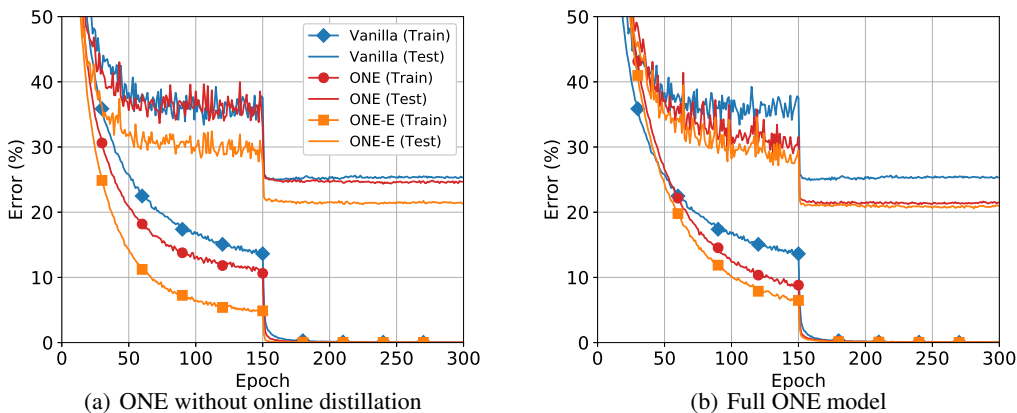

(a) ONE without online distillation          (b) Full ONE model

Figure 3: Effect of online distillation. Network: ResNet-110.

## 4.5 Model Generalisation Analysis

We aim to give insights on why an ONE trained network yields a better generalisation capability. A few previous studies [33, 34] demonstrate that the width of a local optimum is related to the model generalisation. A general understanding is that, the surfaces of training and test error largely mirror to each other and it is favourable to converge the models to broader optima in training. As such, a trained model remains approximately optimal even under small perturbations at test time. Next, we exploited this criterion to examine the quality of model solutions $\boldsymbol{\theta}_v, \boldsymbol{\theta}_m, \boldsymbol{\theta}_o$ discovered by the vanilla, DML and ONE training algorithms respectively. This analysis was conducted on CIFAR100 using ResNet-110.

Specifically, to test the width of local optimum, we added small perturbations to the solutions as $\boldsymbol{\theta}_*(d, \boldsymbol{v}) = \boldsymbol{\theta}_* + d \cdot \boldsymbol{v}$, $* \in \{v, m, o\}$ where $\boldsymbol{v}$ is a uniform distributed direction vector with a unit length, and $d \in [0, 5]$ controls the change magnitude. At each magnitude scale, we further sampled randomly 5 different direction vectors to disturb the solutions. We then tested the robustness of all perturbed models in training and test error rates. The training error was quantified as the cross-entropy measurement between the predicted and ground-truth label distributions.

We observed in Fig 4 that: (1) The robustness of each solution against parameter perturbation appears to indicate the width of local optima as: $\boldsymbol{\theta}_v < \boldsymbol{\theta}_m < \boldsymbol{\theta}_o$. That is, ONE seems to find the widest local minimum among the three therefore more likely to generalise better than others.

(2) Comparing with DML, vanilla and ONE found deeper local optima with lower training errors. This indicates that DML may probably get stuck in training, therefore scarifying the vanilla's exploring capability for more generalisable solutions to exchange the ability of identifying wider optima. In contrast, our method further improves the capability of identifying wider minima over DML whilst maintaining the original exploring quality.

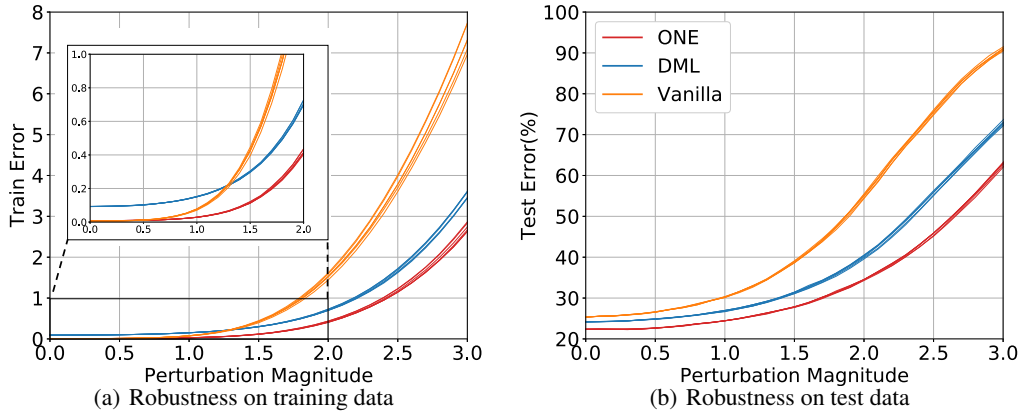

(a) Robustness on training data      (b) Robustness on test data

Figure 4: Robustness test of ResNet-110 solutions found by ONE, DML, and vanilla training algorithms on CIFAR100. Each curve corresponds to a specific perturbation direction $v$.

### 4.6 Variance Analysis on ONE's Branches

We analysed the variance of ONE's branches over the training epochs in comparison to the conventional ensemble method. We used ResNet-32 as the base net and tested CIFAR100. We quantified the model variance by the average prediction differences on training samples between every two models/branches in Euclidean space. Figure 5 shows that a 3-Net Ensemble involves *larger* inter-model variances than ONE with 3 branches throughout the training process. This means that the branches of ONE have higher correlations, due to the proposed learning constraint from the distillation loss that enforces them align to the same teacher prediction, which probably hurts the ensemble performance.

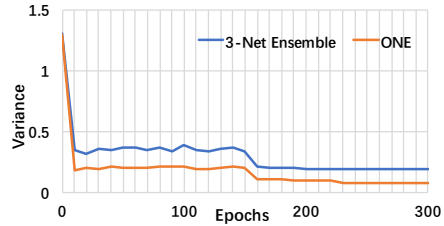

Figure 5: Model variance during training.

However, in the mean generalisation capability (another fundamental aspect in ensemble learning), ONE's branches (the average error rate 26.61±0.06%) are much superior to individual models of a conventional ensemble (31.07±0.41%), leading to a stronger ensembling performance.

## 5 Conclusion

In this work, we presented a novel On-the-fly Native Ensemble (ONE) strategy for improving deep network learning through online knowledge distillation in a one-stage training procedure. With ONE, we can more discriminatively learn both small and large networks with less computational cost, beyond the conventional offline alternatives that are typically formulated to learn better small models alone. Our method is also superior over existing online counterparts due to the unique capability of constructing a high-capacity online teacher to more effectively mine knowledge from the training data and supervise the target network concurrently. Extensive experiments on four image classification benchmarks show that a variety of deep networks can all benefit from the ONE approach. Significantly, smaller networks obtain more performance gains, making our method specially good for low-memory and fast execution scenarios.

## Acknowledgements

This work was partly supported by the China Scholarship Council, Vision Semantics Limited, the Royal Society Newton Advanced Fellowship Programme (NA150459), and Innovate UK Industrial Challenge Project on Developing and Commercialising Intelligent Video Analytics Solutions for Public Safety (98111-571149).

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
