[Reviews · NeurIPS 2018]

Reviewer 1



Summary: Authors propose a novel multi-branch network with a loss function that uses distillation from a combined branch to distill into individual branches. The technique is motivated by the idea that Teacher-Student knowledge distillation is a two-step process often requiring a large pre-trained teacher. Their method builds a teacher, out of weighted ensemble and uses that to train the network. They are able to show that the combined network (ONE-E) is far superior to standalone networks, and the individual branch (ONE) is also better than its counterpart (i.e if it were trained without any of the loss functions and the branches). Pros: 1. Excellent write-up This is a very well written paper. Literature study is comprehensive and the whole paper is easy to follow even if it looks dense in contentwise. 2. Intuitive and excellent idea The idea presented is both intuitive and excellent. People have been using an ensemble of models for various tasks, and it is well known that ensemble will give better results as long as there is some variance in the model. Another approach to improve the accuracy of the model is to train it with soft-logits of a bigger model. They were able to take advantage of both the techniques (ensemble and knowledge distillation) in a single setup. 3. Excellent ablation study They have presented their technique, as a network modification (adding branches, sharing layers) and change is an objective function (KL div loss + CE). They show the result of each of these techniques individually (Table 5) and show that the combination of all of these provides the best result. This leaves no doubt to the reader on the importance of each of the changes. 4. Superior results Their results on CIFAR/SVHN/Imagenet all look very good and promising. It looks like the technique improves the result for various types of model, thus it is agnostic to the structure (Table 1). It outperforms other ensemble techniques (Table 4) which are more computationally expensive than the presented ONE technique. Cons: 1. Section 4.5 feels incomplete and an after-thought. I understand that space was a huge limitation, however, if possible, I would like to see a better analysis of why this works compared to other ensemble and Mixture of Expert techniques. Ensembles work when the models have a variance between them. There is nothing in the current setup to encourage that. Then why does it do better? In a 5 branch setup, how different are these branches? Do they make the same errors or different errors? (Minor) 2. I would like to see more ImageNet results, especially with more efficient models like SENet. 3. How does ONE-E compare to Sparsely Gated MoE [1]? 4. It was not mentioned how many low-level layers where shared. [1] https://arxiv.org/pdf/1701.06538.pdf

Reviewer 2



This paper proposed an online approach to train deep models. The approach trained the deep networks with multiply branches (including the target network) simultaneously. Then the trained single target network or the ensemble could be used for deployment. However, some equations were very confused. For example, Eqn.6 calculated the total loss with i from 0 to m as well as teacher loss. Moreover, the parameter is the same as the original network. Can it considered as an ensemble learning? Some theory analysis should be provided. In addition, the efficiency of the training should be discussed.

Reviewer 3



This paper presents a method for online distillation of deep neural networks. Online distillation methods avoid the complexity of distillation with two-stage training using a larger teacher network. This is done by training multiple base models in parallel and setting a loss function that moves the output of each individual base model toward the aggregate/consensus output. The main idea of this work compared to other online distillation techniques is that the proposed ensemble, base networks share a part of the architecture, and a gating mechanism is used to combine the outputs into aggregate logits. This technique keeps the aggregate network small, making it possible to train the full aggregate model in one stage and without a need for synchronization (compared to [16]). The proposed method is empirically tested against both offline two-stage (KD) and online (DML) distillation methods. The results, somewhat surprisingly, show that the proposed method outperforms the offline method in test accuracy. The required computation is also considerably smaller than the KD method. The paper also shows that obtained final network with the proposed method is in a wider local optima by showing that it is less sensitive to random perturbation of the parameters. Other experiments in the paper show how online distillation and sharing of the layers can act as regularizers when we use the full ensemble at test time. Quality and clarity: the paper is mainly an empirical analysis of a new distillation method. There is no concrete theoretical justification for the proposed method. The paper lightly touches on the idea that the proposed aggregate network architecture is benefiting from regularization effects of sharing network layers and the gating mechanism. Part of the paper are hard to read with a few broken sentences in different sections. Originality and significance: The idea of online distillation is not new (see [15][16]), but the proposal to share layers when co-distilling base networks is interesting and to the best of my knowledge original. Although the idea is marginal, the result seem interesting and significant. Question and comments: How do you decide which layers to share in the aggregate model? Why not take the base network with index = argmax g_i? The choice of scaling the KL loss by T^2 is not well justified. This could be a hyper parameter that you can optimize for with cross validation. Notes: Line 17: necessary -> necessity (maybe just revise the sentence) Line 54-56: Broken sentence Line 70-71: Broken sentence Line 116: While sharing most layers … Line 240: Comparing thetas is not meaningful. Maybe revise the sentence. I have read the authors' feedback.